# Transcriptomic Analysis from Normal Glucose Tolerance to T2D of Obese Individuals Using Bioinformatic Tools

**DOI:** 10.3390/ijms24076337

**Published:** 2023-03-28

**Authors:** Khaoula Errafii, Said Boujraf, Mohamed Chikri

**Affiliations:** 1Clinical Neurosciences Laboratory, Faculty of Medicine and Pharmacy, Sidi Mohammad Ben Abdullah University, Fez 30000, Morocco; 2Biochemistry and Molecular Biology Laboratory, Faculty of Medicine and Pharmacy, Sidi Mohammad Ben Abdullah University, Fez 30000, Morocco; 3African Genome Center, Mohamed IV Polytechnic University, Benguerir 43151, Morocco; 4Inserm Unite CNRS, Lille University UMR 1283-8199, F-59000 Lille, France

**Keywords:** T2D, obesity, insulin resistance, transcriptomics

## Abstract

Understanding the role of white adipose tissue (WAT) in the occurrence and progression of metabolic syndrome is of considerable interest; among the metabolic syndromes are obesity and type 2 diabetes (T2D). Insulin resistance is a key factor in the development of T2D. When the target cells become resistant to insulin, the pancreas responds by producing more insulin to try to lower blood glucose. Over time, this can lead to a state of hyperinsulinemia (high levels of insulin in the blood), which can further exacerbate insulin resistance and contribute to the development of T2D. In order to understand the difference between healthy and unhealthy obese individuals, we have used published transcriptomic profiling to compare differences between the WAT obtained from obese diabetics and subjects who are obese with normal glucose tolerance and insulin resistance. The identification of aberrantly expressed messenger RNA (mRNA) and the resulting molecular interactions and signaling networks is essential for a better understanding of the progression from normal glucose-tolerant obese individuals to obese diabetics. Computational analyses using Ingenuity Pathway Analysis (IPA) identified multiple activated signaling networks in obesity progression from insulin-resistant and normal glucose-tolerant (IR-NGT) individuals to those with T2D. The pathways affected are: Tumor Necrosis Factor (TNF), Extracellular signal-Regulated protein Kinase 1/2 ERK1/2, Interleukin 1 A (IL1A), Protein kinase C (Pkcs), Convertase C5, Vascular endothelial growth factor (Vegf), REL-associated protein (RELA), Interleukin1/1 B (IL1/1B), Triggering receptor expressed on myeloid cells (TREM1) and Nuclear factor KB1 (NFKB1) networks, while functional annotation highlighted Liver X Receptor (LXR) activation, phagosome formation, tumor microenvironment pathway, LPS/IL-1 mediated inhibition of RXR function, TREM1 signaling and IL-6 signaling. Together, by conducting a thorough bioinformatics study of protein-coding RNAs, prospective targets could be exploited to clarify the molecular pathways underlying the development of obesity-related type 2 diabetes.

## 1. Introduction

The worldwide obesity pandemic is driving a significant rise in the incidence of metabolic diseases, such as T2D [1]. Obesity, T2D and insulin resistance are all well-known to be closely linked [2]; diabetes brought on by insulin resistance is triggered by obesity [3]. As a result, it is critical to look at whether obesity is a causal risk factor for T2D in its early stages. In obesity, the body’s adipose tissue secretes proinflammatory cytokines and adipokines, which can impair insulin signaling and promote insulin resistance [4]. This happens due to the activation of several signaling pathways, such as the JNK, IKKβ/NF-κB and SOCS3 pathways, which inhibit insulin signaling and reduce glucose uptake by muscle and adipose cells [5,6]. Furthermore, inflammation has been linked to both obesity and insulin resistance [7]. Diet-induced obesity causes insulin resistance in mouse brown adipose tissue, according to a new study [8]. In mice, natural killer cells have been shown to mediate the link between fat and insulin resistance [9]. Adipose tissue releases more non-esterified fatty acids, glycerol, hormones and proinflammatory cytokines in obese people, which may contribute to the emergence of insulin resistance [10]. The direction of the driving connection between obesity and insulin resistance in humans, however, is yet unknown. Hence the need to understand how obesity directly causes insulin resistance. 

Adipose tissue is one of the most essential tissues contributing to obesity because of its fundamental roles of lipogenesis (fat storage) and lipolysis (fat mobilization) [11]. The primary function of WAT is to store energy in the form of triglycerides and release it when needed [12]. WAT also secretes several hormones and cytokines that regulate energy metabolism, immune function and inflammation [13]. On the other hand, the primary function of brown adipose tissue (BAT) is to generate heat through a process called thermogenesis [14]. This process occurs in specialized organelles called mitochondria, which contain a protein called uncoupling protein 1 (UCP1) that uncouples the electron transport chain from ATP synthesis, resulting in the production of heat instead of ATP [15]. As a result, it is crucial to figure out how much this endocrine tissue plays a role in insulin resistance and T2D. Insulin targets adipose tissue as one of its primary targets besides muscle and liver [16]. In adipocytes, insulin efficiently inhibits triglyceride (TG) breakdown and the release of free fatty acid (FFA) into circulation [17]. In obese people, however, insulin’s suppressive effect on lipolysis is diminished [18]. Insulin resistance in adipose tissue causes extra FFA to be released into circulation, resulting in increased diacylglycerol (DAG) and triacylglycerol (TAG) production in muscle cells and hepatocytes, culminating in ectopic fat deposition [19]. Consequently, DAG can activate protein kinase C (PKC) in muscle and liver [20], inhibiting insulin signaling and promoting skeletal muscle and liver insulin resistance, leading to systemic insulin resistance [21] and a variety of metabolic disorders such as hyperglycemia, hypertension, dyslipidemia, non-alcoholic fatty liver (NAFLD) and T2D [22]. Several studies revealed differentially expressed genes (DEGs) associated to insulin resistance [23]. However, the difference in transcriptomic progression and core pathways affected from normal glucose-tolerant obese individuals to T2D obese individuals in WAT is still unclear. 

The NCBI-Gene Expression Omnibus (GEO) is a public database that contains a high quantity of gene expression data [24], including data from next-generation sequencing [25]. With the flood of gene data generated by high-throughput biology research, identifying and analyzing differentially expressed genes (DEGs) is a crucial yet vast task. However, this data may be analyzed using an integrated bioinformatics approach to find DEGs related to certain human diseases. 

For that purpose, we found recent GEO gene datasets from human adipose tissues related to obesity and T2DM [26]. We examined the expression of genes in human WAT and their relationship with obesity, insulin resistance and T2D to determine the possible pathways affected by dysregulated genes in human WAT and introduce a deeper comprehensive analysis of the progression of the disease. 

## 2. Results

### 2.1. Identification of DEGs in Obese IR-NGT and Obese T2D

The Sequence Read Archive (SRA) database was used to collect expression data for lean individuals, obese individuals with IR-NGT, and obese patients with T2D. The CLC Genomics Workbench 21 program was used to examine each Fastq file and determine the DEGs, followed by a comparison of Gene Expression (GE) tracks of the datasets relating to lean individuals and obese individuals with IR-NGT. The results of hierarchical clustering indicated two distinct DEGs in the obese IR-NGT and obese T2D groups when compared to lean subjects (Figure 1). Heat-map analyses of significant DEGs showed a progressive change of gene expression from lean individuals to obese individuals with IR-NGT and to obese individuals with T2D in WAT. The Venn diagram shows a total of 593 DEGs from obese individuals with IR-NGT were extracted, which were then submitted to differential expression analysis (Figure 2 ), revealing 368 upregulated and 225 downregulated DEGs (absolute(abs) fold change > 2, *p*-value ≥ 0.05; Table 1). In addition, a total of 1474 DEGs from obese patients with T2D were extracted (Figure 2), which was then submitted to differential expression analysis, revealing 1087 upregulated and 387 downregulated DEGs (fold change abs value > 2, *p*-value ≥ 0.05; Table 2). Indeed, both obese groups showed a similar pattern of DEGs when compared to lean individuals, but these changes were most marked in obese individuals with T2D.

### 2.2. Functional Enrichment Analysis Highlights Inhibition of Inflammatory Responses in Obese IR-NGT versus an Activation of Inflammatory Response and Immune Cell Trafficking of WAT in Obese T2D Patients

To get a better understanding of the possible involvement of DEGs in obese IR-NGT pathology and obese T2D in WAT, obese patients with IR-NGT and obese patients with T2D datasets were submitted to IPA and downstream impact analysis (DEA). Affected functional categories are depicted as a heat tree map, which clusters functionally related categories together, resulting in a high-level view of enriched functional categories; the orange boxes represent activated functions, whereas the blue boxes represent suppressed functions. The data shown in (Figure 3) indicates a number of functional categories with negative z-score, such as leukocyte migration, blood cell adhesion, cell movement of monocytes and phagocyte recruitment. Most of the functional categories are categorized under immune cell trafficking, inflammatory response and cellular movement in obese patients with IR-NGT (Figure 4). On the other hand, the group of obese patients with T2D showed overall enrichment of inflammatory response and similar functional categories (Figure 4). These results indicated a crosstalk between the immune system and insulin sensitivity. The dysregulation of different functions in white adipose tissue was most pronounced in obese patients with T2D. Previous studies showed that obesity-associated adipose tissue inflammation is a major cause of decreased insulin sensitivity seen in T2D. 

### 2.3. Multiple Affected Signaling Network in Obese IR-NGT and Obese T2D Compared to Lean Subjects

To identify potential upstream regulators, such as transcription factors (TFs) and any gene or small molecule that has been observed experimentally to affect gene expression by analyzing linkage to DEGs through coordinated expression, we used the new feature in IPA, the upstream regulator analysis (URA) tool. IPA discovered 594 putative upstream regulators that have been detected (Bonferroni adjusted *p*-value 0.05). The upstream regulators most inhibited by negative Z scores were TNF, ERK1/2, IL1A, Pkc(s), C5, Vegf, Oncostatin M (OSM), RELA, IL1/1B, TREM1 and NFKB1, while Nuclear Receptor Subfamily 1 Group H Member 3 (NR1H3), Retinoid X Receptor Alpha RXRA, Cbp/p300-interacting transactivator with Glu/Asp-rich carboxy-terminal domain 2 CITED2 and Nr1h are enriched in the obese patients in the IR-NGT group (Figure 5). The obese patients with T2D showed a direct contrast of upstream regulator expression pattern (Figure 5). This contrast raised the question of what is unified in the pathogenesis of obese patients with T2D and what is individualized. An improved understanding of the relationship between obesity and type 2 diabetes may pave the way for more successful and cost-efficient approaches for both diseases, including more customized treatment. Taking TNF as an upstream regulator found in both groups’ analyses, integrating the results of the TNF-targeted genes and the DEGs in canonical pathways, we found that the dysregulation of TNF-targeted genes (Figure 6), Signal Transducer and Activator of Transcription 1/3 STAT1 and STAT3 contributed to the insulin secretion-signaling pathway. TNF, NFkB complex, NFkBIA, NFkB1 and RELA were also involved in type 2 diabetes signaling. Moreover, HIF1A and TP53 in the adipogenesis pathway were regulated by TNF (Figure 7). 

### 2.4. Identification of Significantly Enriched Canonical Signaling Pathways in Obese IR-NGT and Obese T2D in Comparison with Lean

In order to investigate significantly involved signaling pathways involved in obese individuals with IR and normal glucose tolerance, DEGs from the dataset were subjected to IPA canonical pathways analysis. The activated canonical signaling pathways in obese individuals with IR-NGT were screened after categorizing the identified canonical signaling pathway according to *p*-value and the number of gene sets in each canonical pathway. The top-enriched canonical signaling pathways were assigned to LXR/RXR Activation. On the other hand, phagosome formation, the tumor microenvironment pathway, LPS/IL-1 mediated inhibition of RXR function, TREM1 signaling and IL-6 signaling were categorized with negative z-score (Figure 8). Interestingly, among the 167 significant canonical pathways identified by IPA (*p*-value 0.05), the following had absolute z-scores greater than 2.0, phagosome formation, CREB signaling in neurons, cardiac hypertrophy signaling, neuroinflammation signaling pathway and TREM signaling were enriched, indicating that they were strongly associated to T2D in the obese individuals with T2D (Figure 8). Unbiased IPA analysis identified several canonical pathway terms related to cellular immune response, cytokine signaling, lipid metabolism and immune cell trafficking; these results confirm the progression of WAT inflammation from lean subjects to obese subjects with IR-NGT and obese subjects with T2D and indicate that upregulation in the inflammatory response is most pronounced in the group with T2D. 

## 3. Discussion

Excess white adipose tissue (WAT) mass has been related to a variety of problems, including insulin resistance, type 2 diabetes and obesity [27]. Recent studies suggested that reduced insulin sensitivity in adipocytes is a precursor to impaired WAT function and whole-body insulin resistance [28]. Insulin has a complicated effect on its target cells by activating multiple pathways that regulate glucose and lipid metabolism [29]. Understanding the underlying molecular mechanisms might be a fresh approach to identifying prospective targets suffering from obesity and T2D and provide more tailored therapy. 

The primary goal of this project was to identify adipocyte-specific mRNAs linked to obesity, insulin resistance and type 2 diabetes. We used 25 RNA-seq samples to characterize gene expression profiles and identify significant transcriptional regulators in this study. To avoid methodological inconsistencies in terms of data processing and bioinformatics pipelines among original research, we used a consistent bioinformatics workflow to handle the raw RNA-seq data (fastq files) of all datasets. As a result, we provide a transcriptome analysis study of white adipose tissue from lean subjects, obese subjects with IR-NGT and obese subjects with T2D. 

According to the results of the IPA pathway enrichment analysis, significant DEGs in obese subjects with IR-NGT were primarily involved in the activation of LXR/RXR activation, while the same pathway was inhibited in obese subjects with T2D. Liver X Receptors are integrators of metabolic and inflammatory signaling. LXRs play a central role in lipid metabolism regulation. In accordance with our results, previous studies showed that the activation of LXRs improves glucose tolerance in adipose tissue in the murine model [30].

Additionally, for obese subjects with IR-NGT, the inhibited pathways were primarily involved in phagosome formation, TREM signaling, IL-6/8/17 signaling and leukocyte extravasation signaling. On the other hand, obese subjects with T2D showed a contrasting pattern of dysregulated canonical pathways. The latter group showed significant enrichment in phagosome formation and IL-6/8/17 signaling. These dysregulated pathways lie within the cellular immune response [31,32,33]. Recent studies showed increasing evidence of T2D’s impact on the immune system; the immune response to elevated blood glucose causes an inflammatory reaction along with inflammatory mediators released by adipocytes [34].

Studies have well established that there is a link between obesity and T2D; however, not all obese patients develop T2D [35]. The precise mechanisms linking the conditions remain unclear, as does our comprehension of inter-individual differences. Previous studies have also identified the possible link between obesity and T2D [36]. This link has been shown to involve proinflammatory cytokines such as the tumor necrosis factor and interleukin-6 [37]. In our study, TNFs were significantly downregulated in obese subjects with IR-NGT and significantly upregulated in obese subjects with T2D. 

Lipid metabolism functions in Adipose tissue is a simple surrogate measure that represents pathophysiological changes in adipose tissue insulin sensitivity, with increasing change from normal weight to obesity and from NGT to type 2 diabetes [38]. In our study, we found that lipid metabolism functions such as adipose tissue lipolysis, the concentration of phospholipids, lipid flux, the concentration of cholesterol, synthesis of triacylglycerol and lipid efflux have higher Z-scores and are enriched in T2D while they are downregulated in obese subjects with IR-NGT. FA concentration, lipid conversion, lipid concentration and FA metabolism are enriched in obese subjects with IR-NGT when compared to those with T2D. These data indicate a divergence in the lipid metabolism function analysis. 

## 4. Materials and Methods

### 4.1. Source of Data

We searched the National Center of Biotechnology Information (NCBI) GEO database (https://www.ncbi.nlm.nih.gov/geo/ (accessed on 15 May 2022)) using the keywords “Type 2 diabetes”, “obesity”, “insulin resistance” and “normal glucose tolerance”. We found one dataset (GSE141432) that described two studies: Measurements of adipose tissue lipolysis by microdialysis, a piece of case–control research examining the causes of obesity-driven insulin resistance, and adipose tissue genetics. Figure 9 provides specific facts.

### 4.2. Next-Generation Sequencing Data

We utilized a GSE141432 set gene expression profile. It was downloaded from the GEO database, and it was based on the NextSeq 500 platform using the Illumina TruSeq Total Stranded Ribo Zero library preparation and was submitted by Fryk et al. [26]. We extracted the GSE141432 dataset containing 9 lean subjects, 7 obese insulin-resistant and normal glucose-tolerant (IR-NGT) subjects and 9 obese subjects with Type 2 Diabetes (T2D). The CLC Genomics Workbench Version 21.0.4. Software (QIAGEN, Hilden, Germany) was used to look for differences in gene expression between lean individuals, obese individuals with normal glucose tolerance with insulin resistance and obese individuals with type 2 diabetes from white adipose tissue. After the retrieval of raw RNA sequencing data from the sequence read archive (SRA) database CLC Genomics Workbench-21 was used to align pair-end reads to the hg38 human reference genome. TPM (Transcript Per Kilobase Million) mapped reads were used to calculate the amount of transcript expression. An ANOVA test was performed among the three groups to detect DEGs with an absolute value of the 2-fold change. A statistically significant difference was defined as a *p*-value that is equal to or less than 0.05. Figure 10 describes the methodological approach followed; however, the data of lncRNAs (long non-coding RNAs) are still under investigation.

### 4.3. IPA Pathway Enrichment Analysis of DEGs 

To discover DEGs at the biologically functional level, Ingenuity Pathway Analysis IPA (Ingenuity Systems; www.ingenuity.com/, accessed on 20 January 2022) was used. Upstream regulator analysis (URA), downstream effects analysis (DEA), mechanistic networks (MN) and causal network analysis (CNA) prediction algorithms were used to perform functional annotations and regulatory network analysis [39]. IPA predicts functional regulatory networks from gene expression data using a precise algorithm and assigns a significance score to each network based on its fit to the set of focus genes in the database. The *p*-value is the negative log of P and indicates the likelihood that the network’s focal genes were discovered together by coincidence [40].

## 5. Conclusions

The present study utilized bioinformatic tools to perform a comprehensive investigation of differentially expressed genes (DEGs) that may play a role in the development of insulin resistance, obesity with normal glucose tolerance (NGT) and type 2 diabetes (T2D). The results obtained from this analysis offer a significant contribution to our understanding of the molecular mechanisms that underlie these complex metabolic disorders. The identification of these DEGs provides new avenues for future research, which may lead to the discovery of novel therapeutic targets. Despite the potential significance of these findings, further molecular biological tests are needed to confirm the specific roles of the identified genes. It is essential to validate these findings in experimental models, including cell cultures and animal models, to confirm the functional significance of these genes. Such studies may provide more detailed insights into the molecular mechanisms underlying insulin resistance and obesity with NGT and T2D and may ultimately lead to the development of more effective treatments for these disorders. Overall, the findings of this study highlight the potential of bioinformatic tools for identifying key genes involved in the development of metabolic diseases. However, it is important to recognize that these findings are preliminary and must be further validated through additional experimental investigations. With further research, it may be possible to develop targeted therapies that address the underlying molecular processes involved in these complex metabolic disorders.

## Figures and Tables

**Figure 1 ijms-24-06337-f001:**
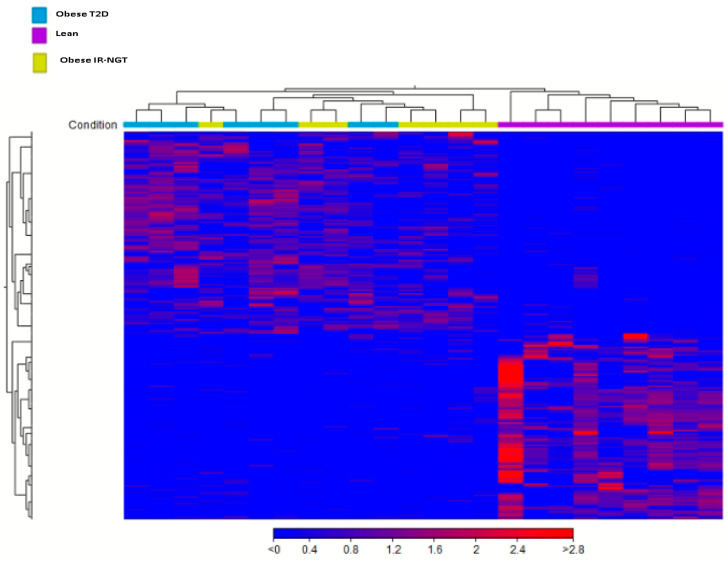
Cluster analysis of most significant DEGs. The abscissa represents different samples n = 25; the vertical axis represents clusters of DEGs. Red color represents upregulation; blue color represents downregulation. Expression data are represented as normalized values. Legend shows the clustering of sample categories purple: lean/control, green: obese T2D, yellow: Obese IR-NGT.

**Figure 2 ijms-24-06337-f002:**
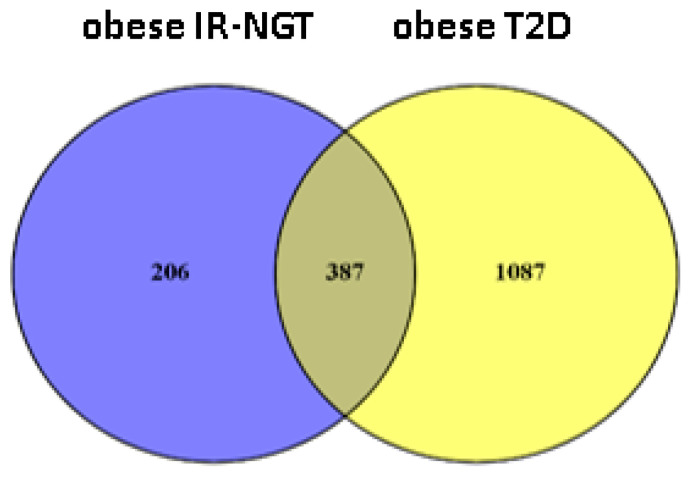
Venn diagram of differential expression analysis DEGs of a comparison between both obese IR-NGT and obese T2D groups. Yellow color represents obese T2D vs. lean, and the blue color represents Obese IR-NGT vs. lean.

**Figure 3 ijms-24-06337-f003:**
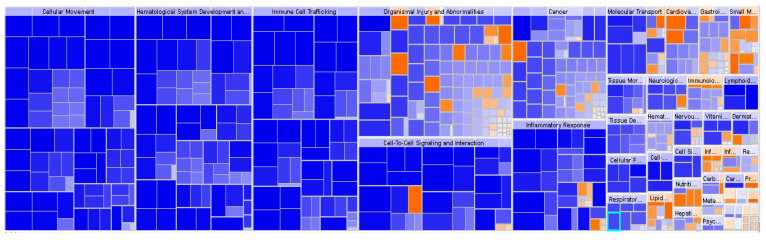
Functional category annotation analysis of obese individuals in IR-NGT group, when compared to lean group, shows inhibition of cellular movement, immune cell trafficking and cell-to-cell signaling and interaction. Blue color indicates inhibition; red color indicates activation.

**Figure 4 ijms-24-06337-f004:**
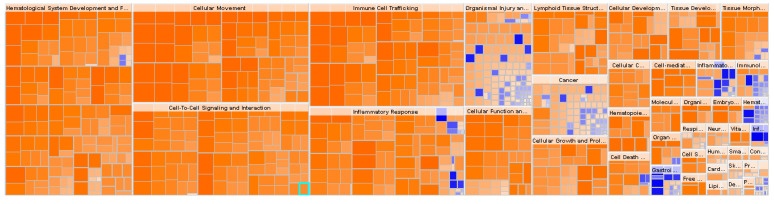
Functional category annotation analysis of obese patients with T2D shows a significant enrichment of hematological system development and function, cellular movement, immune cell trafficking and cell-to-cell signaling and interaction, inflammatory response and cellular functions and maintenance. Blue color indicates inhibition; red color indicates activation.

**Figure 5 ijms-24-06337-f005:**
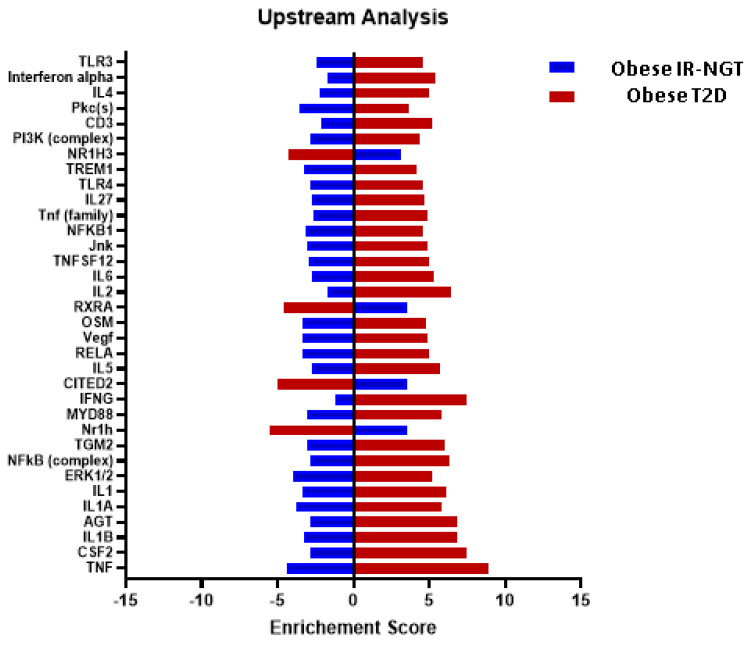
Upstream regulator analysis of differentially expressed genes in obese patients with IR-NGT and obese patients with T2D. Y-axis indicates the upstream regulator, and the x-axis represents the activation Z score. Red bars represent obese individuals with T2D, and blue bars represent obese individuals with IR-NGT.

**Figure 6 ijms-24-06337-f006:**
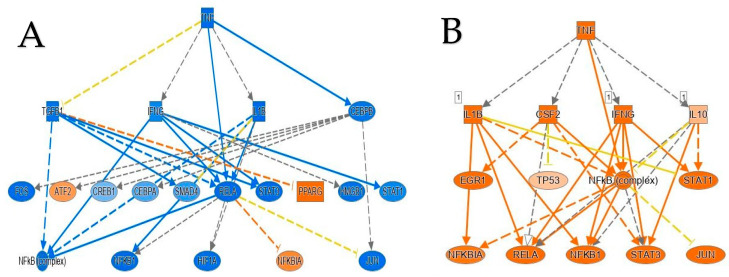
Regulator affects network analysis based on IPA highlighting a role for activated TNF-affected signaling networks comparison. (**A**) This figure is representation of TNF downregulated network according to IPA prediction in obese individuals with IR-NGT. (**B**) TNF is enriched as an upstream regulator network in obese individuals with T2D compared to lean individuals based on IPA analysis.

**Figure 7 ijms-24-06337-f007:**
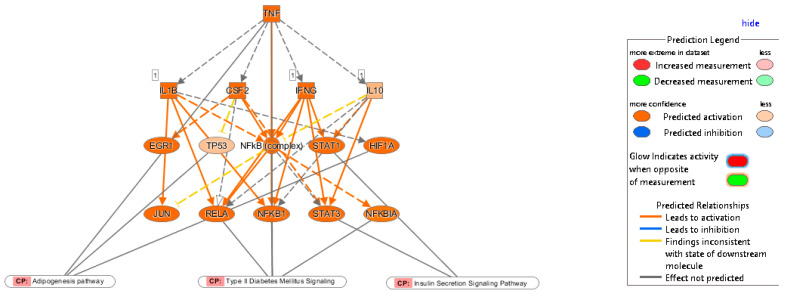
Network of TNF in obese subjects with T2D, its targeted genes and corresponding T2D. For example, activation of TNF leads to upregulation (indicated by orange line) of DEG (shown by orange color). Upregulation of DEGs further affects insulin secretion signaling pathway (CP). For other indicators, please refer to the prediction legend.

**Figure 8 ijms-24-06337-f008:**
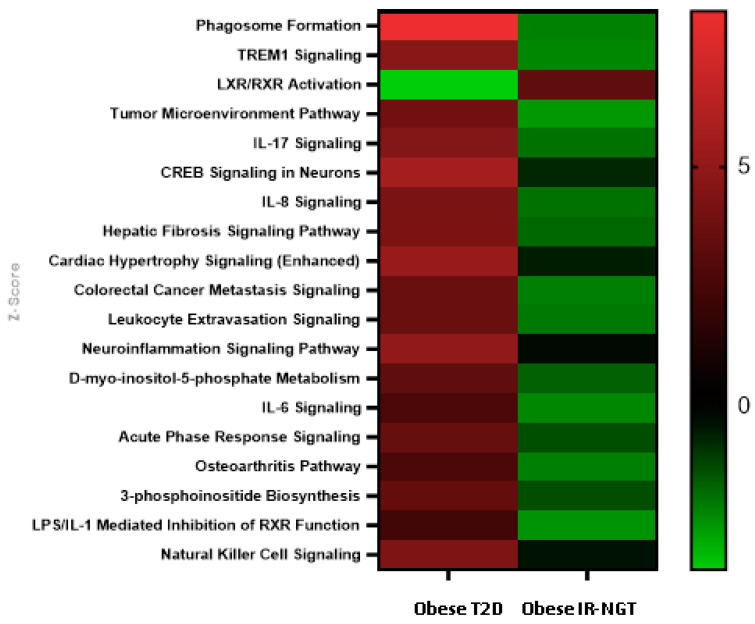
Top dysregulated canonical signaling pathways. The log (*p*-value), z-score and ratio of top significantly activated canonical signaling pathways are indicated in red color. A scale from light green to dark red indicates the level of inhibition/activation in/of the canonical signaling pathways, respectively.

**Figure 9 ijms-24-06337-f009:**
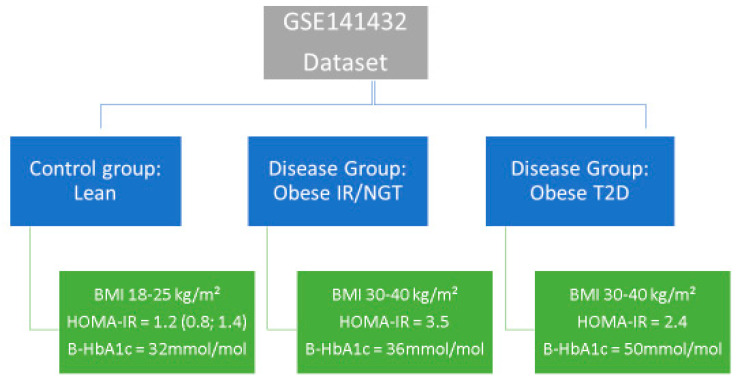
Dataset baseline information.

**Figure 10 ijms-24-06337-f010:**
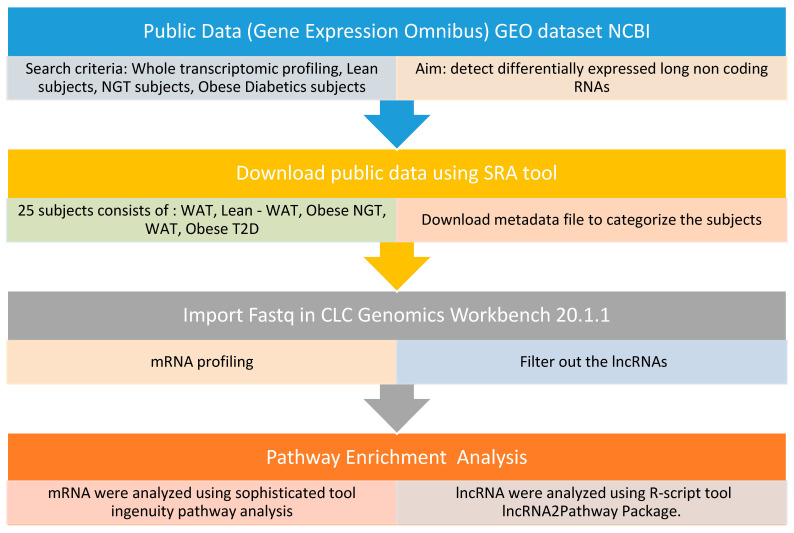
Bioinformatics analysis workflow from the retrieval of the public data of GEO to the analysis of dysregulated/affected pathways.

**Table 1 ijms-24-06337-t001:** Top 20 DEGs in obese insulin resistant—normal glucose tolerance.

Top Upregulated DEGs Obese IR-NGT			Top Downregulated DEGs Obese IR-NGT		
Name	Log_2_ Fold Change	Description	Name	Log_2_ Fold Change	Description
RNF17	5.66	ring finger protein 17	DCD	−7.77	dermcidin
CTAG2	4.83	cancer/testis antigen 2	MUCL1	−6.66	mucin-like 1
TBC1D3K	4.69	TBC1 domain family member 3K	SCGB2A2	−6.66	secretoglobin family 2A member 2
TBC1D3E	4.13	TBC1 domain family member 3E	SCGB1D2	−6.19	secretoglobin family 1D member 2
DEUP1	3.95	deuterosome assembly protein 1	MSLN	−5.09	mesothelin
TMEM215	3.77	transmembrane protein 215	BORCS7-ASMT	−4.76	BORCS7-ASMT readthrough (NMD candidate)
CEACAM20	3.66	CEA cell adhesion molecule 20	NTSR2	−4.66	neurotensin receptor 2
C2orf83	3.42	chromosome 2 open reading frame 83	CA6	−4.66	carbonic anhydrase 6
DPYSL4	3.37	dihydropyrimidinase-like 4	EEF1E1-BLOC1S5	−4.08	EEF1E1-BLOC1S5 readthrough (NMD candidate)
ZNF723	3.24	zinc finger protein 723	ZPBP	−3.78	zona pellucida binding protein
PMCH	3.23	pro-melanin concentrating hormone	ABHD16B	−3.57	abhydrolase domain containing 16B
RPP21	3.2	ribonuclease P/MRP subunit p21	GABRR3	−3.53	gamma-aminobutyric acid type A receptor subunit rho3
TRIM39-RPP21	3.15	TRIM39-RPP21 readthrough	RASSF6	−3.38	Ras association domain family member 6
COMP	3.14	cartilage oligomeric matrix protein	SPX	−3.27	spexin hormone
HOXD10	3.07	homeobox D10	IQCA1L	−3.24	IQ motif containing AAA domain 1-like
CCDC54	3.06	coiled-coil domain containing 54	CYP1A2	−3.12	cytochrome P450 family 1 subfamily A member 2

**Table 2 ijms-24-06337-t002:** Top 20 DEGs in obese T2D.

Top Upregulated DEGs Obese T2D			Top Downregulated DEGs Obese T2D		
Name	Log_2_ Fold Change	Description	Name	Log_2_ Fold Change	Description
HBG2	5.14	hemoglobin subunit gamma 2	TMEM52	−1.05	transmembrane protein 52
AC008763.3	5.04	alpha hemoglobin stabilizing protein	CFAP74	−2.84	cilia and flagella associated protein 74
HBD	4.45	hemoglobin subunit delta	GABRD	−3.07	gamma-aminobutyric acid type A receptor subunit delta
DEFA1B	4.1	defensin alpha 1B	ARHGEF16	−1.26	Rho guanine nucleotide exchange factor 16
AC139530.2	4.07	5′-aminolevulinate synthase 2	TAS1R1	−1.11	taste 1 receptor member 1
ARMH2	3.8	armadillo-like helical domain containing 2	CA6	−2.62	carbonic anhydrase 6
MAG	3.7	myelin-associated glycoprotein	SLC25A34	−1.4	solute carrier family 25 member 34
TMEM215	3.62	transmembrane protein 215	CLCNKB	−1.31	chloride voltage-gated channel Kb
AC104389.6	3.62	prokineticin 2	PLA2G5	−1.05	phospholipase A2 group V
PTPRN	3.59	protein tyrosine phosphatase receptor type N	GRIK3	−1.53	glutamate ionotropic receptor kainate type subunit 3
FAM72C	3.53	family with sequence similarity 72 member C	GJA9	−1.87	gap junction protein alpha 9
POPDC3	3.53	popeye domain containing 3	NT5C1A	−1.34	5′-nucleotidase, cytosolic IA
LCN1	3.53	lipocalin 1	HYI	−1.12	hydroxypyruvate isomerase (putative)
NPFF	3.53	neuropeptide FF-amide peptide precursor	AGBL4	−1.47	ATP/GTP binding protein-like 4
HBA1	3.51	hemoglobin subunit alpha 2	ELAVL4	−1.25	ELAV-like RNA binding protein 4
IZUMO3	3.48	IZUMO family member 3	CDKN2C	−1.62	cyclin-dependent kinase inhibitor 2C
LY6G6F	3.42	lymphocyte antigen 6 family member G6F	GLIS1	−1.41	GLIS family zinc finger 1
EGFL6	3.38	EGF-like domain multiple 6	FOXD3	−1.77	forkhead box D3
CAMP	3.37	cathelicidin antimicrobial peptide	TTLL7	−1.1	tubulin tyrosine ligase-like 7
HBB	3.37	hemoglobin subunit beta	MCOLN3	−1.01	mucolipin TRP cation channel 3
AC034102.2	3.37	chromosome 2 open reading frame 83	GBP7	−2.33	guanylate-binding protein 7
ITLN1	3.29	intelectin 1	UBL4B	−1.3	ubiquitin-like 4B
COMP	3.21	cartilage oligomeric matrix protein	CHIA	−2.71	chitinase acidic
GYPB	3.2	glycophorin B (MNS blood group)	CASQ2	−1.77	calsequestrin 2
SLC4A1	3.19	solute carrier family 4 member 1 (Diego blood group)	PHGDH	−1.49	phosphoglycerate dehydrogenase
TRIM10	3.18	tripartite motif containing 10	CIART	−1.01	circadian-associated repressor of transcription
HBQ1	3.17	hemoglobin subunit theta 1	RORC	−1.43	RAR-related orphan receptor C
DUSP13	3.16	dual specificity phosphatase 13	S100A1	−1.74	S100 calcium-binding protein A1
KLF1	3.13	Kruppel-like factor 1	NUP210L	−1.13	nucleoporin 210-like
KCNA10	3.13	potassium voltage-gated channel subfamily A member 10	DCST2	−1.03	DC-STAMP domain containing 2
JCHAIN	3.1	joining chain of multimeric IgA and IgM	NHLH1	−2.31	nescient helix-loop-helix 1
CMTM2	3.1	CKLF-like MARVEL transmembrane domain containing 2	TSTD1	−1.09	thiosulfate sulfurtransferase-like domain containing 1
CA1	3.07	carbonic anhydrase 1	SPATA46	−1.57	spermatogenesis associated 46
KRT72	3.05	keratin 72	FAM78B	−1.41	family with sequence similarity 78 member B
HBM	3.05	hemoglobin subunit mu	MAEL	−2.1	maelstrom spermatogenic transposon silencer
S100P	3.01	S100 calcium-binding protein P	SLC19A2	−1.05	solute carrier family 19 member 2
MMP7	3.01	matrix metallopeptidase 7	AXDND1	−1.4	axonemal dynein light-chain domain containing 1
RIPPLY2	3.01	ripply transcriptional repressor 2	GLUL	−1.53	glutamate-ammonia ligase
CBLIF	3.01	cobalamin binding intrinsic factor	ADORA1	−1.22	adenosine A1 receptor
TMEM132D	3	transmembrane protein 132D	LEFTY2	−1.91	left-right determination factor 2
ZNF723	2.99	zinc finger protein 723	COQ8A	−1.1	coenzyme Q8A
KLRC4	2.99	killer cell lectin-like receptor C4	TRIM67	−1.35	Novel protein
FCGR3B	2.98	Fc fragment of IgG receptor IIIb	NTSR2	−4.63	tripartite motif containing 67
IFIT1B	2.97	interferon-induced protein with tetratricopeptide repeats 1B	LPIN1	−1.59	neurotensin receptor 2
FUT7	2.96	fucosyltransferase 7	VSNL1	−1.79	lipin 1
S100A8	2.95	S100 calcium-binding protein A8	APOB	−2.26	visinin-like 1

## Data Availability

The data are available under project number PRJNA593449.

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
