# Peer review of "Transcriptomic Analysis from Normal Glucose Tolerance to T2D of Obese Individuals Using Bioinformatic Tools"

_ijms, 2023, doi:10.3390/ijms24076337_

Round 1
Reviewer 1 Report
Dear Authors,
You undertook an interesting task of transcriptomic analysis from normal glucose tolerance to T2D of obese individuals using bioinformatics tools. For this purpose, a number of bioinformatics tools were used to draw preliminary conclusions.
Unfortunately, I was a bit disappointed in the way the results were presented, and the methodology is not written in a way that allows you to reproduce the results.
It is necessary to revise the manuscript for publication due to listed below reasons.
Major concerns:
Due to the multi-stage procedure, I recommend presenting materials and methods in the form of graphics. Please, if possible, try to fill in the manuscrypt with table or graphic, showing what the different stages of data selection and standardization look like. Please be very precise in materials and methods section.
Maybe it is worth highlighting which markers in your opinion are the most important and why?
Please expand the conclusions section.
Minor concerns:
- 1. Acronyms must be written out in full the first time they are used.
- 2. The use of capital letters is inconsistent and not always justified, e.g. verse 25 and “Signaling” but line 24 “activation”.
- Line 27: did you mean “…obesity-T2D pathogenesis e.g. can be identified”?
- Line 50: what did you mean in “Several studies revealed insulin resistance differentially expressed genes”. I am not sure if the work has undergone proper language proofreading, there are some misspelled sentences and strange constructions.
- How was the patient selection process? how was it known that the obese control was "healthy"? How many groups were there? Obese NGT, Obese T2D, lean, obese IR-NGT.
- Line 69: what is GE in “following comparing GE tracks of Lean 69 and Obese IR NGT datasets.”
- Line 72 and others: what means fold change Abs (abs) – capital letters are inconsistent.
- Line 101-103 – very long and confusing subtitle
- Line 133 and others: abbreviations must be expanded hen used first time in the text
- I will recommend better presentation of results. How patients were selected? How many groups you have analyzed? Data are very interesting but yous should present them better.
- I have problem because once you compare Obese IR and Obese T2D. Did you have any control? Than you are writing about collecting data for lean, obese IR with NGT and Obese T2D patients. Please make it more clear which data were compared with what and what the selection looked like.
Discussion:
- Please check typos and grammar.
Kind regards,
Reviewer 2 Report
This test analyzed and identified the differential expression mRNA (DEGs) between white adipose tissue of obese diabetes patients and obese normal glucose tolerance subjects, and used IPA analysis to find that the obesity progression from NGT individuals to T2D individuals activated multiple signal networks including TNF, ERK1/2, IL1A, Pkc, etc; The functional annotation focused on LXR activation, Phagosome formation, Tumor Microenvironment Pathway, LPS/IL-1 Mediated Inhibition of RXR Function, TREM1 Signaling and IL-6 Signaling, which confirmed the existence of crosstalk between the immune system and insulin sensitivity, that is, the inflammatory reaction of T2D white adipose tissue is the main reason for the decrease of insulin sensitivity. The test design is reasonable, the content is detailed, and the test method is feasible. However, the published data is used for IPA analysis in this paper. The data sampling may have regional, age, body mass index, sampling position and other differences, which ultimately leads to the reduction of the reliability of the results. If the sampling range of the data is defined or the samples are roughly classified, the test integrity and the representativeness of the results will be better; In addition, the NGT group designed in this trial includes thin NGT, normal NGT and obese IR-NGT, but the description of the test results seems to focus only on the difference between IR-NGT and obese T2D. If the thin NTG and normal NTG can be described slightly, the test integrity and result representativeness will be better. The thesis writing is relatively smooth, with clear basic ideas and clear logic, but there are still some problems that need to be corrected:
1. In the abstract, the relationship between insulin resistance and T2D is not connected, which is inconsistent with the research purpose;
2. In the abstract, NTG and T2D, when abbreviations appear for the first time, the full name should be indicated first;
3. It is recommended to improve the content of the introduction. For example, the relationship between obesity and insulin resistance has been mentioned many times in the article, but it is not specific enough. It is recommended to review more relevant literature, especially the research results on signal pathways, and provide more argumentative support, so as to facilitate the correspondence with the pathway network of the results of this paper; In addition, should the introduction add the reason why white adipose tissue is selected for research, such as the functional difference between white adipose tissue and brown adipose tissue? How does it relate to metabolic diseases? There is also a lack of statement on IPA analysis.
4. In the results, it is recommended to check the normalization of the full text chart, for example, "Table 2" should be changed to "Table 2."; The definition of legends such as Figure 6 needs to be adjusted; The annotation and legend name should be consistent, as shown in Figure 1: Obesity IR-NGT and obesity NGT, and the name should also be consistent in the text;
5. In the experimental design, the description of the sample range division of published data is not clear enough. Is it to analyze "published data in recent years" or "all published data"? Is the sample affected by region, age, body mass index, and the sampling site of white fat? In addition, the test only carried out IPA analysis on the data and lacked the verification of the accuracy of the results.
6. In the discussion, row 235 , test design and results did not reflect DELs.
Round 2
Reviewer 1 Report
Dear Authors,
In my opinion you the paper has been well improved.
I can submit to the editor's decision a paper that is worth publishing in the International Journal of Molecular Sciences.
Kind regads,